# RETHINKING THE ACTOR-CRITIC NETWORKS USING HYBRID QUANTUM-CLASSICAL PARADIGM

## ABSTRACT

We present a novel hybrid quantum-classical actor-critic reinforcement learning (RL) model. In the noisy intermediate-scale quantum (NISQ) era, full utilization of qubits is impractical due to resource limitations. To tackle this issue, this paper proposes Quantum-Critic Proximal Policy Optimization (QC-PPO), where the critic is designed using Quantum Neural Networks, whereas an actor is implemented using conventional networks. We further argue that allocating quantum capacity to the critic serves as a more natural lever for performance gains in actor-critic RL. This is because bootstrapped value estimates shape advantage computation, which consequently shapes the direction of every policy update. Evaluations on multiple MuJoCo environments show consistent improvements; on Humanoid-v4, QC-PPO improves the median return by 52.3% over PPO with equal environment steps, demonstrating its potential for on-board applications.

## 1 INTRODUCTION

Quantum reinforcement learning (QRL) replaces the deep neural network (DNN) with a quantum neural network (QNN; parameterized quantum circuit, PQC). Prior works have reported performance gains and parameter efficiency (fewer learnable parameters) gains on simple low-dimensional environments (e.g., CartPole, MountainCar) (Chen et al., 2020; Jerbi et al., 2021; Cho et al., 2024). However, as the number of qubits and circuit depths increases, gradients vanish due to the *barren plateau* (McClean et al., 2018), posing challenges for training. Constraints from the noisy intermediate-scale quantum (NISQ) era and limited qubit budgets further restrict scaling of PQC. As a result, the scaling of state–action dimensionality is restricted and observed advantages mostly appear in low-dimensional environments. In practice, the lack of quantum inference hardware and the high computational cost of simulators hinder practical deployment.

As shown by Kölle et al. (2024); Jin et al. (2025), hybrid deep quantum neural networks (hDQNNs), which place a PQC between classical DNN modules, improve training stability and expressivity, mitigating optimization difficulties associated with *barren plateaus* and partially addressing state–action dimensionality. While they partially address state-action dimensionality, parameter-shift–based updates (Wierichs et al., 2022) require many PQC evaluations. This creates a training bottleneck that also burdens gradient calculation of the pre-PQC deep neural network (PreDNN). Gradient surrogates such as quantum tangential deep neural network (qtDNN) enable end-to-end backpropagation in hDQNNs by providing learned gradients for the PQC block (Luo & Chen, 2025). This study reports that combining hDQNN with qtDNN alleviated the PQC evaluation bottleneck and thus improved scalability to higher-dimensional tasks. Nevertheless, most prior work focuses on quantum actors, which require quantum resources at deployment and are impractical for on-board systems such as robotics and autonomous platforms.

In this paper, we concentrate quantum capacity in the critic. In actor–critic methods, update directions are largely determined by the critic (Yang et al., 2022). Moreover, under long horizons or large discount factors, value functions accumulate high-frequency components, whereas standard multi-layer perceptrons (MLPs) exhibit spectral bias (Yang et al., 2022) toward low frequencies. PQCs, via data re-uploading and entanglement, can systematically expand the accessible frequency spectrum and have been reported to capture higher frequencies more effectively (Xu & Zhang, 2024).

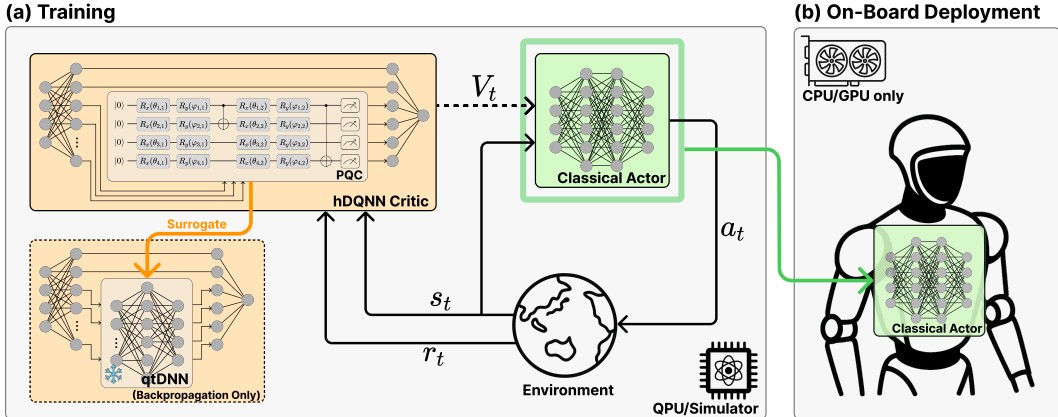

Figure 1: Overview of QC-PPO. **(a)** Training. We replace the PPO critic with a hybrid deep QNN (hDQNN; *PreDNN→PQC→PostDNN*). The forward pass goes through the hDQNN to produce value estimates, while gradients are routed through a qtDNN surrogate used for backpropagation only, enabling scalable training in high-dimensional state–action spaces. **(b)** On-board deployment. At deployment, only the classical actor runs on a central processing unit (CPU) or a GPU—no quantum hardware or simulator is required.

Therefore, we propose QC-PPO, which replaces the proximal policy optimization (PPO) critic with an hDQNN while keeping the actor purely classical. The PQC block is trained via a qtDNN surrogate, enabling efficient end-to-end critic updates on classical accelerators such as graphic processing units (GPUs). During training, the critic's PQC forward passes run on a quantum processing unit (QPU) or a simulator while the qtDNN surrogate supplies gradients for backpropagation on classical hardware. At deployment, only the classical actor runs on-board, eliminating the need for quantum hardware and ensuring low-latency inference.

The key contributions can be summarized as follows. First, we empirically show that PPO performance is more sensitive to critic expressivity than to the actor expressivity; on Humanoid-v4, under equal conditions, QC-PPO improves the best-median return by about 52.3% over an MLP-based PPO baseline (Table 1). Second, we propose QC-PPO, which achieves train–deploy separation with a hybrid critic and a classical actor, meeting on-board latency requirements; empirically, the actor's inference is approximately $3.4\times$ faster than a quantum actor on the same GPU (Table 3). Third, we propose error-aware annealed gradient blending (EAGB) for qtDNN (Luo & Chen, 2025), which increases the weight of surrogate gradients as the qtDNN error decreases(Section 4.3); this improves early-training stability while reducing the number of quantum evaluations. To facilitate reproducibility, we will release our training/evaluation scripts upon acceptance.

## 2 RELATED WORK

PQCs have recently been studied as learnable models that leverage qubit operations and provide a different inductive bias from classical networks. Applied to RL, Chen et al. (2020); Jerbi et al. (2021); Cho et al. (2024) reported that PQC-based models could achieve higher performance with fewer parameters, especially in low-dimensional settings. However, most results remained confined to such settings due to *barren plateau* problems and the high cost of training.

To address these limitations, hybrid models that combine PQCs with DNNs have been proposed. In these architectures, classical networks handle data pre- and post-processing, while the PQC performs high-dimensional feature extraction, thereby improving overall model expressivity and helping to mitigate optimization difficulties associated with *barren plateaus* (Kölle et al., 2024; Jin et al., 2025). In particular, most of the architectures implement quantum actors that use quantum circuits to generate more diverse action trials during exploration. Although some environments report performance gains, the practicality of quantum actors remains limited by challenges in on-board deployment and by long training times. We therefore propose a practical algorithm that addresses the limited maturity of quantum-specific optimization and mitigates key training bottlenecks.

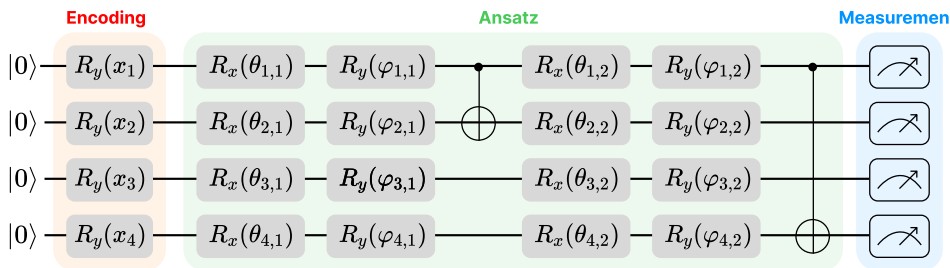

Figure 2: PQC overview. Inputs $x$ are encoded by $U_{\text{enc}}(x)$, processed by $U_{\text{ans}}(\boldsymbol{\theta})$, and measured via $B$ to obtain $f_{\boldsymbol{\theta}}(x)$; gradients are obtained with the parameter-shift rule.

Several ablation studies have moved the quantum module to the critic, in many cases, this configuration outperforms a quantum actor (Jin et al., 2025; Kölle et al., 2024). These findings were typically presented without a clear explanation, and sometimes attributed to hyperparameter choices or tuning artifacts. In contrast, we argue that the effect is systematic: actor–critic performance is more sensitive to the critic's expressivity, and a hybrid critic (hDQNN) directly mitigates an expressivity bottleneck in the value function. We substantiate this hypothesis with spectral analysis of the learned value function and corresponding performance gains (Figure 3).

## 3 BACKGROUND

**Parameterized Quantum Circuit.** A PQC is a learnable quantum circuit that combines data encoding, a parameterized *ansatz*, and measurements. A single qubit is $|\psi\rangle = \alpha |0\rangle + \beta |1\rangle$ with $|\alpha|^2 + |\beta|^2 = 1$ and equivalently written as the column vector $|\psi\rangle = \begin{pmatrix} \alpha \\ \beta \end{pmatrix}$. Let $X$, $Y$, and $Z$ denote the Pauli matrices; single-qubit rotations about the Bloch axes are $R_x(\theta) = e^{-i\frac{\theta}{2}X}$, $R_y(\varphi) = e^{-i\frac{\varphi}{2}Y}$, and $R_z(\phi) = e^{-i\frac{\phi}{2}Z}$, while two-qubit entangling gates such as controlled-NOT (CNOT) and controlled-Z (CZ) create correlations among qubits. Classical inputs $x$ are embedded via data-encoding rotations (e.g., angle encoding with $R_{x/y/z}$), followed by a parameterized ansatz $U_{\text{ans}}(\boldsymbol{\theta})$; the overall unitary is $U(x) = U_{\text{ans}}(\boldsymbol{\theta}) U_{\text{enc}}(x)$. Measuring an observable $B$ yields the model output, as follows,

$$f_{\boldsymbol{\theta}}(x) = \langle\psi| U^{\dagger}(x) B U(x) |\psi\rangle, \tag{1}$$

and its expectation is estimated with $N_{\text{shots}}$ repeated measurements. The gradient of the expectation with respect to $\theta_k$ can be obtained without differentiating through the simulator via the parameter-shift rule,

$$\frac{\partial f_{\boldsymbol{\theta}}(x)}{\partial \theta_k} = \frac{1}{2}\Big[ f_{\boldsymbol{\theta_k}+\frac{\pi}{2}}(x) - f_{\boldsymbol{\theta_k}-\frac{\pi}{2}}(x) \Big], \tag{2}$$

thus, a single gradient component requires two forward circuit evaluations. For a given input–observable pair, the circuit-call cost therefore scales as follows,

$$2 \times N_{\text{params}} \times N_{\text{shots}}, \tag{3}$$

and this evaluation burden compounds with circuit depth and the number of measured observables. Beyond the well-known *barren plateau* phenomenon, the parameter-shift cost and shot noise introduce an additional training bottleneck that limits scalability in high-dimensional settings.

**Quantum Tangential Deep Neural Network.** *Parameter-shift* rule (Wierichs et al., 2022) based parameter updates in PQCs require multiple forward circuit evaluations per parameter, which in practice calls for high-throughput quantum execution (ideally in parallel). In reality, the limited availability of QPUs that support such parallelism constrains the practicality and scalability of PQCs.

The qtDNN addresses this by training a surrogate model of the PQC within a quantum–classical hybrid pipeline. During backpropagation, it bridges gradients between the PreDNN and the post-PQC deep neural network (PostDNN), thereby alleviating the backpropagation bottleneck through the PQC and the PreDNN.

Furthermore, the proposed hDQNN differs from conventional quantum–classical hybrids in two key respects. First of all, the output of the PreDNN directly parameterizes the PQC's control parameters, i.e., rotation angles and entangling operations; and then, secondly, to exploit both classical and quantum representational capacity, a direct connection link $d_{c-link}$ connects the PreDNN to the PostDNN, providing a classical communication path that can bypass the PQC when beneficial.

Although the architecture introduces an additional surrogate-model training stage, parallel back-propagation parameter updates in classical GPU enabled by qtDNN alleviated the training bottle-necks in the PQC and PreDNN components that were observed in the original hDQNN.

## 4 QUANTUM CRITIC PPO

In this section, we introduce QC-PPO, which instantiates the critic with an hDQNN and the actor with a standard DNN within an actor-critic framework.

### 4.1 MOTIVATION

It has been shown that injecting Fourier features into the MLPs of deep Q-network (DQN) and deep deterministic policy gradient (DDPG) improves performance over plain MLPs Brellmann et al. (2023); Evmorfos et al. (2023). In particular, Yang et al. (2022) demonstrate from a neural tangent kernel (NTK) perspective that the low-frequency bias of value-function MLPs can hinder actor–critic learning.

Building on two observations—(i) most gains arise when *critic* features receive Fourier structure, and (ii) the expectation value of a PQC under a single-parameter gate $U(x) = e^{-ixG}$ admits a Fourier-series expansion with frequencies drawn from eigenvalue differences of $G$—we hypothesize that a PQC-based critic can realize a similar advantage:

$$f_Q(x) = \sum_{\omega \in \Omega} c_\omega e^{i\omega x} = a_0 + \sum_{\ell=1}^{R} \big[a_\ell \cos(\Omega_\ell x) + b_\ell \sin(\Omega_\ell x)\big], \tag{4}$$

with $\Omega \subseteq \{\lambda_j - \lambda_k\}$ for eigenvalues $\{\lambda_j\}$ of $G$.

While Luo & Chen (2025) applied hDQNN/qtDNN to the twindelayed deep deterministic policy gradient algorithm (TD3) *actor*, we instead place the hybrid module in the *critic* to directly address limited value-function expressivity. We adopt PPO for its clipped updates and parallel training efficiency—crucial in safety-critical control where rapid, diverse interaction is preferable to maximal sample reuse. During training, a qtDNN surrogate carries gradients through the quantum block, avoiding parameter-shift overhead.

### 4.2 THE HDQNN CRITIC UPDATE WITH QTDNN

We retain the PPO pipeline but replace the *critic* with an hDQNN. In this setup, the PQC itself has no free internal parameters; instead, the PreDNN outputs a control vector $q_i$ that directly parameterizes the PQC's rotation gates. Thus, trainability comes indirectly through the PreDNN weights.

The critic maps $s \mapsto V(s)$ as follows,

$$q_i, d_{c\text{-}link} = \text{PreDNN}(s), \tag{5}$$

$$q_o = f_Q(q_i) \equiv \text{PQC}(q_i), \tag{6}$$

$$V(s) = \text{PostDNN}([q_o, d_{c\text{-}link}]), \tag{7}$$

*where* $s \in \mathbb{R}^{d_s}$; $q_i \in \mathbb{R}^{N_{\text{params}}}$ is the PreDNN control vector; $d_{c\text{-}link} \in \mathbb{R}^{N_{d_{c\text{-}link}}}$ is a classical bypass; $f_Q$ denotes the non-trainable PQC; $q_o \in [-1, 1]^{n_q}$ stacks Pauli-$Z$ expectations; and the concatenation $[q_o, d_{c\text{-}link}]$ is fed into PostDNN to produce the scalar value $V(s)$.

During rollouts we store PQC inputs and outputs $(q_i, q_o)$ in a buffer. From these, tiny batches $B_{\text{tiny}}$ are drawn to train a qtDNN surrogate:

$$\mathcal{L}_{\text{dist}} = \frac{1}{B_{\text{tiny}}} \sum_{i=1}^{B_{\text{tiny}}} \|f_{\text{qt}}(q_i; \omega) - f_Q(q_i)\|_2^2. \tag{8}$$

---

**Algorithm 1** Pseudocode for the proposed QC-PPO algorithm.

---

1: Initialize policy $\pi_\theta$; critic PreDNN and PostDNN; qtDNN surrogate $f_{\mathrm{qt}}(\cdot; \omega)$
2: **for** iteration $= 1, 2, \dots$ **do**
3:     **for** $t = 1$ to $T$ **do**
4:         Sample $a_t \sim \pi_\theta(\cdot \mid s_t)$; step env to get $(r_t, s_{t+1})$
5:         $(q_{i,t}, d_{c\text{-}link,t}) \leftarrow \mathrm{PreDNN}(s_t)$
6:         $q_{o,t} \leftarrow f_Q(q_{i,t})$                  ▷ PQC output: Pauli-$Z$ expectations
7:         $V_t \leftarrow \mathrm{PostDNN}([\,q_{o,t},\, d_{c\text{-}link,t}\,])$
8:         Store $(s_t, a_t, r_t, V_t, q_{i,t}, q_{o,t}, d_{c\text{-}link,t})$
9:     **end for**
10:    Compute GAE advantages $\hat{A}_t$ and targets $\hat{R}_t = \hat{A}_t + V_t$
11:    **Distill qtDNN (buffer from rollout):**

$$\mathcal{L}_{\mathrm{dist}} = \tfrac{1}{B} \sum_{i=1}^{B} \left\| f_{\mathrm{qt}}(q_i; \omega) - f_Q(q_i) \right\|_2^2, \quad \omega \leftarrow \omega - \eta \nabla_\omega \mathcal{L}_{\mathrm{dist}}$$

12:    **EAGB schedule:**

$$\alpha_t = \begin{cases} 0, & \text{if iteration} \leq T_{\mathrm{warm}}, \\ \alpha^\star \cdot \min\!\left(1, \dfrac{\text{iteration} - T_{\mathrm{warm}}}{T_{\mathrm{ramp}}}\right), & \text{otherwise.} \end{cases}$$

13:    **for** epoch $= 1$ to $K$ **do**
14:       **for** minibatch $\mathcal{B}$ **do**
15:         Policy ratio $r_t = \dfrac{\pi_\theta(a_t \mid s_t)}{\pi_{\theta_{\mathrm{old}}}(a_t \mid s_t)}$
16:         **EAGB (critic forward):**

$$\tilde{q}_{o,t} = (1 - \alpha_t)\, f_{\mathrm{qt}}(q_{i,t})^{\mathrm{detach}} + \alpha_t\, f_{\mathrm{qt}}(q_{i,t}),$$

$$V_t = \mathrm{PostDNN}([\,\tilde{q}_{o,t},\, d_{c\text{-}link,t}\,])$$

17:         **Actor loss:**

$$\mathcal{L}_{\mathrm{actor}} = -\mathbb{E}_{\mathcal{B}}\Big[\min\big(r_t \hat{A}_t,\ \mathrm{clip}(r_t, 1 \pm \varepsilon)\, \hat{A}_t\big)\Big]$$

18:         **Critic loss:**   $\mathcal{L}_{\mathrm{critic}} = \mathbb{E}_{\mathcal{B}}\big[(V_t - \hat{R}_t)^2\big]$   (value clipping optional)
19:         **Total loss:**   $\mathcal{L} = \mathcal{L}_{\mathrm{actor}} + c_v\, \mathcal{L}_{\mathrm{critic}}$
20:         Update $\theta$ and critic params by Adam on $\nabla \mathcal{L}$; optionally update $\omega$ on $\nabla \mathcal{L}_{\mathrm{dist}}$
21:       **end for**
22:    **end for**
23: **end for**

---

*where* $f_{\mathrm{qt}}(\cdot; \omega)$ is the qtDNN surrogate with weights $\omega$; $B_{\mathrm{tiny}}$ is the distillation mini-batch size.

Since Pauli-$Z$ expectation outputs are continuous expectation values, mean squared error (MSE) is more natural than cross-entropy, which is tailored to probabilistic targets and can cause gradient blow-up near the boundaries. The trained qtDNN then replaces the PQC during backpropagation, providing classical differentiability. Consistency of qtDNN gradients has been analyzed in Luo & Chen (2025).

## 4.3 ERROR-AWARE ANNEALED GRADIENT BLENDING

Early in training, surrogate gradients can be unreliable. We introduce *error-aware annealed gradient blending (EAGB)*: initially updates rely on the always-available classical path ($d_{c-link}$), while qtDNN influence is increased gradually. Concretely,

$$\tilde{f}_{\mathrm{qt}}(q_i; \alpha) = (1 - \alpha)\, f_{\mathrm{qt}}^{\mathrm{detach}}(q_i) + \alpha\, f_{\mathrm{qt}}(q_i), \tag{9}$$

where $f_{\mathrm{qt}}^{\mathrm{detach}}$ is a stop-gradient copy ($\partial f_{\mathrm{qt}}^{\mathrm{detach}}/\partial \omega = 0$). The qtDNN weights $\omega$ are still trained by $\mathcal{L}_{\mathrm{dist}}$.

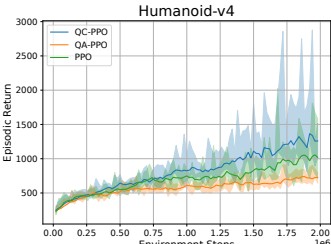 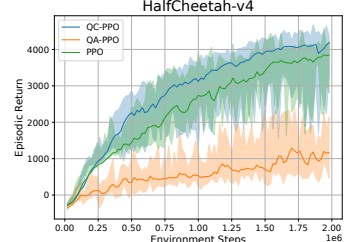 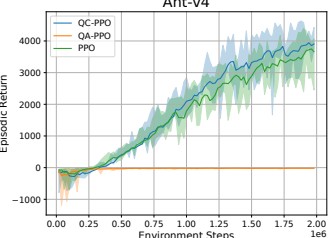

Figure 3: Median EMA learning curves on Humanoid-v4, HalfCheetah-v4 and Ant-v4 (IQR shaded, 10 seeds).

We use a linear ramp schedule

$$\alpha_t = \alpha^\star \cdot \min\Big(1, \frac{\max(0,\, t - T_{\mathrm{warm}})}{T_{\mathrm{ramp}}}\Big), \tag{10}$$

with target $\alpha^\star \in (0, 1]$. Optionally, "error-aware" variants tie the increase of $\alpha_t$ to the surrogate calibration error $\|f_{\mathrm{qt}} - f_Q\|^2$, enabling adaptive gating.

### 4.4 ALGORITHM SUMMARY

Our proposed framework is summarized in Algorithm 1. In (Line 1), we initialize the policy $\pi_\theta$, the critic submodules (PreDNN and PostDNN), the qtDNN surrogate $f_{\mathrm{qt}}(\cdot; \omega)$. In (Lines 2–9), we collect a rollout of length $T$: at each step we sample $a_t \sim \pi_\theta(\cdot \mid s_t)$, compute $(q_{i,t}, d_{c\text{-}link,t}) = \mathrm{PreDNN}(s_t)$, evaluate the PQC; $q_{o,t} = f_Q(q_{i,t})$ (Pauli-$Z$ expectations), obtain the value $V_t = \mathrm{PostDNN}([\,q_{o,t}, d_{c\text{-}link,t}\,])$, and store all tuples for learning. In (Line 10), we compute GAE advantages $\hat{A}_t$ and value targets. In (Lines 11–12), we distill the qtDNN on buffered pairs $(q_i, q_o)$ via the MSE loss $\mathcal{L}_{\mathrm{dist}}$ and set the EAGB blending coefficient $\alpha_t$ using the warm-up/ramp schedule. In (Lines 13–21), we update parameters over $K$ epochs and mini-batches: we form the EAGB-blended critic input, compute the value estimation, optimize the clipped PPO actor loss and the MSE critic loss, and apply Adam updates to $\theta$ and the critic parameters.

## 5 EVALUATION

We evaluate our architecture on multiple high-dimensional MuJoCo environments including the challenging Humanoid-v4 benchmark (Brockman et al., 2016). Because learning curves in these environments exhibit substantial variability across random seeds—even for the same algorithm (Mania et al., 2018)—we conduct a multi-seed evaluation.

**Protocols.** For each method (ours and baselines), we run experiments with 10 random seeds and report the median and interquartile range (IQR; 25–75%) of episodic return across seeds. To align sample budgets across runs, trajectories are resampled onto a common step grid with stride 20,000 via linear interpolation, and per-step percentiles are computed on this grid. For visualization, we optionally apply an EMA-smoothed median curve, but all statistical comparisons (tables and numerical results) are based on the unsmoothed medians and IQRs.

**Baselines and fairness.** To isolate the role of a quantum critic, we evaluate three configurations.
1. QC-PPO (ours): quantum-augmented critic (hDQNN/qtDNN) with a classical actor,
2. QA-PPO: reversed configuration—quantum-augmented actor with a classical critic,
3. PPO: fully classical actor–critic.

For fairness, network sizes are matched across methods (see Table 2). Learning curves are shown in Figure 3.

**Summary metrics.** From each interpolated median curve we report (i) the *peak median*—the maximum median return across training—with its $q_{25}$ and $q_{75}$ at the peak step, and (ii) the *final-at-budget*

Table 1: Median return and IQR ($q_{25}$, median, $q_{75}$) over 10 seeds on three MuJoCo environments.

| Environment | Method | Peak median | | | Final at Budget | | |
| --- | --- | --- | --- | --- | --- | --- | --- |
| | | $q_{25}$ | median | $q_{75}$ | $q_{25}$ | median | $q_{75}$ |
| Humanoid-v4 | QC-PPO | 1277.993 | 1769.126 | 2034.736 | 1011.587 | 1265.081 | 1513.972 |
| | QA-PPO | 641.229 | 782.631 | 844.339 | 643.687 | 717.436 | 795.387 |
| | PPO | 709.735 | 1161.744 | 2024.238 | 719.805 | 946.759 | 1582.493 |
| HalfCheetah-v4 | QC-PPO | 2245.759 | 4342.132 | 4646.062 | 3304.476 | 4297.952 | 4377.111 |
| | QA-PPO | 910.293 | 1626.839 | 2280.093 | 467.834 | 1178.669 | 2146.152 |
| | PPO | 2223.479 | 3968.225 | 4573.391 | 2803.185 | 3850.272 | 4213.181 |
| Ant-v4 | QC-PPO | 3019.571 | 4152.033 | 4630.525 | 3756.357 | 4013.574 | 4429.250 |
| | QA-PPO | -25.461 | -17.073 | -15.966 | -21.053 | -19.727 | -15.787 |
| | PPO | 3420.916 | 3955.076 | 4316.747 | 2449.911 | 3481.606 | 4068.357 |

Table 2: Network architectures. In hDQNN models, the critic and actor are split into PreDNN and PostDNN. The PQC has no trainable parameters, with each layer applying $R_x$ and $R_y$, therefore $N_{\text{params}} = 2 \times N_{\text{qubits}} \times N_{\text{layers}}$ On Humanoid-v4, we use 10 qubits and 10 layers.

| Method | Actor layers | Critic layers |
| --- | --- | --- |
| PPO | $[d_s \rightarrow 512 \rightarrow 128 \rightarrow d_a]$ | $[d_s \rightarrow 256 \rightarrow 256 \rightarrow 1]$ |
| QC-PPO | $[d_s \rightarrow 512 \rightarrow 128 \rightarrow d_a]$ | Pre: $[d_s \rightarrow 256 \rightarrow N_{\textbf{d\text{-}link}} + N_{\textbf{params}}]$
Post: $[N_{\textbf{d\text{-}link}} + N_{\textbf{qubits}} \rightarrow 128 \rightarrow 1]$ |
| QA-PPO | Pre: $[d_s \rightarrow 256 \rightarrow N_{\textbf{d\text{-}link}} + N_{\textbf{params}}]$
Post: $[N_{\textbf{d\text{-}link}} + N_{\textbf{qubits}} \rightarrow 128 \rightarrow d_a]$ | $[d_s \rightarrow 256 \rightarrow 256 \rightarrow 1]$ |

median with its $q_{25}$ and $q_{75}$ (Table 1). This jointly captures transient best performance and end-of-training performance.

**Results.** Across 10 seeds, we observe the performance superiority in the order of

$$\text{QA-PPO} \; < \; \text{PPO} \; < \; \text{QC-PPO},$$

especially with QC-PPO achieving a 52.3% higher best-median return than the classical PPO baseline (Table 1). This supports our hypothesis that allocating quantum capacity to the *critic* more directly enhances value-function expressivity and stabilizes policy updates. By contrast, improvements on the other environments are modest. We hypothesize that this is because a single configuration, with network sizes and hyperparameters tuned primarily for Humanoid-v4, was applied across tasks.

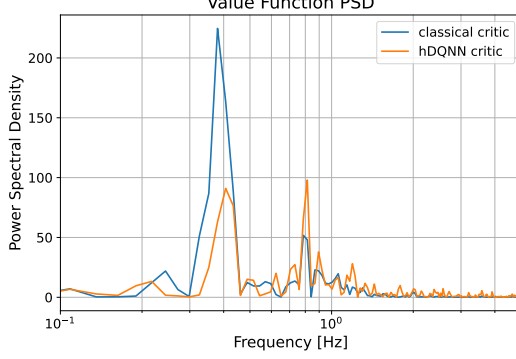

Figure 4: Power spectral density of the learned value function $V(s)$ computed from a single trajectory obtained by concatenating 50 episodes in Humanoid-v4.

Table 3: Actor inference latency (ms; mean $\pm$ standard deviation over 5 runs)

|  | Classical Actor | Quantum Actor |
|---|---|---|
| Time | $0.786 \pm 0.042$ | $2.695 \pm 0.008$ |

**Frequency analysis of the value function.** To compare the high-frequency modeling ability of the critic, we compute the power spectral density (PSD) of the value predictions $V(s_t)$. For fairness, we roll out the same set of evaluation trajectories and feed the resulting states to two critics—the classical critic and our hybrid hDQNN critic. Figure 4 shows that both critics concentrate energy at low frequencies, but the classical critic exhibits a pronounced narrow-band peak around $10^0$ and a heavier concentration near that band. In contrast, the hDQNN critic redistributes energy across multiple moderate bands and reduces narrow-band dominance, yielding a smoother value trace with modest mid–to–high-frequency components. Because advantages $A_t$ depend on temporal differences of $V(s_t)$, the hDQNN's shift in spectral bias likely reduces the variance of $A_t$ and stabilizes policy updates, helping to explain the higher performance achieved by QC-PPO.

**Deployment efficiency.** Finally, we compare actor inference latency between QC-PPO (classical actor) and QA-PPO (quantum actor). Using the same GPU, we measure the mean latency over five runs with CUDA and CUDA-Q (quantum simulator). As shown in Table 3, the classical actor in QC-PPO is approximately $3.4\times$ faster than the quantum actor in QA-PPO. This highlights a practical advantage of our design: quantum resources are required only during training, while deployment remains fully classical.

# 6 CONCLUSION

QC-PPO showed promising performance improvements over a classical MLP-based baseline in the challenging Humanoid-v4 environment, while maintaining the practical advantage of on-board deployment with a purely classical actor. These results position QC-PPO as one of the most practical QRL architectures available today, enabling a clear train–deploy separation in which quantum resources are used only during learning. A limitation of this study is that all experiments were conducted on a CUDA-Q–based GPU quantum simulator rather than real quantum hardware. As future work, we will evaluate training efficiency on QPUs and deploy QC-PPO for policy learning on humanoid robots to validate its real-world applicability.

# LLM USAGE

To improve fluency and clarity, we used a large language model (LLM) for limited editing (e.g., phrasing and grammar). All changes were reviewed by the authors to ensure faithfulness to the intended meaning, and no identifying information was disclosed.

# ETHICS STATEMENT

This work adheres to the ICLR Code of Ethics. The study does not involve human subjects, personally identifiable information, or sensitive attributes. All datasets used are publicly available and employed under their respective licenses. We took care to avoid harmful or biased use of the models.

# REPRODUCIBILITY STATEMENT

We provide sufficient details to reproduce our results, including the model architectures, training hyperparameters, random seeds, and evaluation protocols. Where applicable, we include links or detailed instructions for data preparation and code execution in the supplementary material.

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
