# OpenReview forum: "Rethinking the Actor-Critic Networks using Hybrid Quantum-Classical Paradigm"
_ICLR.cc/2026/Conference — ICLR 2026 Conference Withdrawn Submission_

### Official Review · Reviewer_iskw · 2025-10-31

**Soundness:** 2
**Presentation:** 2
**Contribution:** 2
**Rating:** 4
**Confidence:** 4

**Summary:**

The authors present a hybrid quantum-classical actor-critic reinforcement learning model called Quantum-Critic Proximal Policy Optimization (QC-PPO), where the classical neural network in the critic is replaced by a quantum neural network. QC-PPO is evaluated against QA-PPO (a variant of their model where the actor network is a quantum neural network) and the classical PPO algorithm. The experiments demonstrate a 52.3% improvement in median return over PPO.

**Strengths:**

The paper addresses an interesting and innovative idea by modifying only the critic network in PPO. Furthermore, the idea of training the quantum critic via a quantum tangential deep neural network (qtDNN) surrogate, which enables efficient end-to-end critic updates on classical accelerators such as graphics processing units (GPUs), is particularly interesting and shows promise.

**Weaknesses:**

In the experiments section, the authors show that QC-PPO outperforms both QA-PPO and PPO. However, the reported episodic returns in Figure 3 and Table 1 appear to be significantly lower compared to other studies on the same benchmarks. For example, prior works such as https://arxiv.org/html/2507.18883v1 and https://openreview.net/pdf/7ddae316e3756b236e4454bd720f5843d89fbf7e.pdf report episodic returns around 5000 for Humanoid-v4, whereas the values presented here are much lower. Did the authors employ a different set of hyperparameters, or is there another explanation for the observed performance gap? Further clarification on this issue is required.

Minor Issues:
Page 1: The statement, “Quantum reinforcement learning (QRL) replaces the deep neural network (DNN) with a quantum neural network (QNN; parameterized quantum circuit, PQC),” is not entirely accurate, as this describes only one type of QRL. Other approaches to QRL exist and should be acknowledged.
Page 1: The phrase “limited qubit budgets” is awkward. The authors could revise it to “limited number of qubits” for better readability and clarity.
Page 1: The citation, “As shown by Kölle et al. (2024); Jin et al. (2025),” should be revised to “As shown by Kölle et al. (2024) and Jin et al. (2025)” for consistency and correctness.
Page 2: Similarly, “Chen et al. (2020); Jerbi et al. (2021); Cho et al. (2024)” should be rewritten as “Chen et al. (2020), Jerbi et al. (2021), and Cho et al. (2024).”

**Questions:**

What is the explanation for the observed performance gap compared to prior works on the same benchmarks?

---

### Official Review · Reviewer_Fexf · 2025-10-31

**Soundness:** 2
**Presentation:** 4
**Contribution:** 2
**Rating:** 2
**Confidence:** 4

**Summary:**

This paper introduces a variation on the PPO RL algorithm using a hybrid quantum classic neural network to represent the value function.  The gradients for the quantum part of the network are obtained using a classical surrogate model, which in turn is trained to replicate the quantum input-output behavior. A simulated annealing type of gradient blending is used in order to initially limit the influence of the surrogate model. For evaluation, three continuous control MuJoCo environments are used - including Humanoid-v4 wich features high dimensional state- and action-spaces. The proposed method is shown to outperform classical PPO and PPO with both actor and critic hybrid quantum networks. Finally, a power spectral density analysis is performed on the predicted values of classical and hybrid quantum value functions.

**Strengths:**

The proposed method is a more practicable version of PPO that utilizes hybrid quantum neural networks. It can be applied in a wider range of settings since no quantum capabilities are necessary for the policy. An adequate set of continuous action environments is used for evaluating the method. The approach is well motivated and its explanation is well written. Moreover, the results look promising.

**Weaknesses:**

The paper evaluates the proposed method just against a classical version of the same algorithm. Including other algorithms and especially more recent PQC based approaches - such as HDQNN-TD3 - would improve the paper immensely.
Some of the evaluation details seem questionable (see Questions). Hyperparameter tuning is mentioned but no details or results are presented. An ablation on the presented methods is mostly missing: EAGB with vs. without, parameter-shift rule vs. qtDNN.
The overall contribution appears to be minor since existing methods are combined similarly to another recent paper that did this for TD3.

**Questions:**

- Can you comment on how much improvement stems from the Fourier structure inductive bias that comes from the use of PQCs? Can we use this finding in PPO with classical DNNs?
- What is the wall clock time of training with the proposed algorithm compared to the baseline PPO?
- How does the computational complexity compare to classical and other PQC based approaches?
- What are the actual layer sizes for QC/A-PPO in table 2?
- You say that "To align sample budgets across runs, trajectories are resampled onto a common step grid with stride 20,000 via linear interpolation, and per-step percentiles are computed on this grid". Can you explain this in more detail?
- Your statement "By contrast, improvements on the other environments are modest." comes without specifying before what the tuned-for environment is. This makes me question: How did you tune for this environment (Humanoid)? Did you also tune the baseline accordingly?
- For the Frequency analysis of the value function, you rolled out a single batch of evaluation trajectories and compute the power spectral density for each critic. Which actor did you use for creating the trajectories? (Note that PPO learns a state-value function that is conditioned on the actor.)
- Can you include some more information about the consistency of qtDNN gradients?

---

### Official Review · Reviewer_aPnL · 2025-10-31

**Soundness:** 2
**Presentation:** 2
**Contribution:** 2
**Rating:** 2
**Confidence:** 4

**Summary:**

This paper proposes a variant of hybrid quantum reinforcement learning in which the critic rather than the actor is replaced by a quantum-hybrid network. The motivation is that quantum circuits can provide higher spectral expressivity, potentially improving value estimation in continuous-control domains. The authors present results on three continuous-control benchmarks, showing modest improvements over both classical PPO and a hybrid PPO variant with a quantum-hybrid actor.

While the integration of quantum components into RL is well motivated, the paper’s empirical gains are minor and limited in scope. The method appears to introduce significant computational overhead, while the performance improvements may be attributable to regularization effects or a more expressive classical surrogate rather than genuine advantage from using quantum computing.

**Strengths:**

- The integration of quantum computing concepts into reinforcement learning is well motivated and thoughtfully presented.
- The authors demonstrate awareness of current limitations in hybrid quantum–classical RL and attempt to address expressivity bottlenecks in the critic.
- The spectral-expressivity motivation is interesting and could inspire future work on structured non-classical regularization in RL.

**Weaknesses:**

- Limited novelty: The architectural change (quantum critic instead of actor) is incremental.
- Lack of theoretical foundation: No analysis or intuition beyond spectral expressivity is provided to explain why the proposed critic should yield better value estimation.
- Narrow empirical scope: Evaluation is restricted to three continuous-control environments, all using PPO as baseline. No comparison with other actor–critic variants (e.g., SAC) or multi-critic architectures.
- Weak empirical evidence: Reported gains are small and often within variance; in the humanoid task, even the best-performing models show non-convergent behaviors.
- High computational cost: Simulating PQCs and training an additional surrogate critic add substantial overhead, making practical benefits doubtful.
- Ambiguity of “hybrid” claim: The architecture uses classical pre-/post-processing and a classical surrogate for an untrainable PQC, simulated noise-free—better described as “quantum-inspired” rather than hybrid.
- Presentation issues: Some figures add little information; the paper under-uses available page space; explanations around Fig. 4 do not clearly match the plotted data.

**Questions:**

1. What motivates using an untrainable PQC? How is its structure determined, and how does it interact with the classical surrogate?
2. Why choose a hybrid critic with classical pre-/post-processing instead of a simpler trainable VQC? Would this not offer a fairer comparison?
3. How frequently is the PQC bypassed beyond early training phases, and how does this affect learning dynamics?
4. What are the details and parameter counts of the hDQNN surrogate? Shouldn’t an equivalent architecture be added to the classical baseline for fair comparison?
5. In Table 1, why is the peak median typically higher than the final median? Does this reflect instability or overfitting?

---

### Note · Authors · 2025-11-13

I have read and agree with the venue's withdrawal policy on behalf of myself and my co-authors.